# Thymocyte Development of Humanized Mice Is Promoted by Interactions with Human-Derived Antigen Presenting Cells upon Immunization

**DOI:** 10.3390/ijms241411705

**Published:** 2023-07-20

**Authors:** Takataro Fukuhara, Yoshihiro Ueda, Sung-Il Lee, Tokifumi Odaka, Shinsuke Nakajima, Jun-Ichi Fujisawa, Kazu Okuma, Makoto Naganuma, Kazuichi Okazaki, Naoyuki Kondo, Yuji Kamioka, Mitsuru Matsumoto, Tatsuo Kinashi

**Affiliations:** 1Division of Gastroenterology and Hepatology, The Third Department of Internal Medicine, Kansai Medical University, Hirakata 573-1010, Osaka, Japan; fukuhart@kuzuha.kmu.ac.jp (T.F.); naganuma@hirakata.kmu.ac.jp (M.N.); okazaki@kouri.kmu.ac.jp (K.O.); 2Department of Molecular Genetics, Institute of Biomedical Science, Kansai Medical University, Hirakata 573-1010, Osaka, Japan; kondonao@hirakata.kmu.ac.jp (N.K.); kamiokay@hirakata.kmu.ac.jp (Y.K.); 3Department of Model Animal, Institute of Biomedical Science, Kansai Medical University, Hirakata 573-1010, Osaka, Japan; silee@hirakata.kmu.ac.jp; 4Department of Microbiology, Kansai Medical University, Hirakata 573-1010, Osaka, Japannakajims@hirakata.kmu.ac.jp (S.N.); fujisawa@hirakata.kmu.ac.jp (J.-I.F.); okumak@hirakata.kmu.ac.jp (K.O.); 5Division of Molecular Immunology, Institute for Enzyme Research, Tokushima University, Kuramoto 770-8503, Tokushima, Japan; mitsuru@tokushima-u.ac.jp

**Keywords:** humanized mice, thymus development, MHC, dendritic cells, immunization

## Abstract

Immune responses in humanized mice are generally inefficient without co-transplantation of human thymus or HLA transgenes. Previously, we generated humanized mice via the intra-bone marrow injection of CD133+ cord blood cells into irradiated adult immunodeficient mice (IBMI-huNSG mice), which could mount functional immune responses against HTLV-1, although the underlying mechanisms were still unknown. Here, we investigated thymocyte development in IBMI-huNSG mice, focusing on the roles of human and mouse MHC restriction. IBMI-huNSG mice had normal developmental profiles but aberrant thymic structures. Surprisingly, the thymic medulla-like regions expanded after immunization due to enhanced thymocyte expansion in association with the increase in HLA-DR+ cells, including CD205^+^ dendritic cells (DCs). The organ culture of thymus from immunized IBMI-huNSG mice with a neutralizing antibody to HLA-DR showed the HLA-DR-dependent expansion of CD4 single positive thymocytes. Mature peripheral T-cells exhibited alloreactive proliferation when co-cultured with human peripheral blood mononuclear cells. Live imaging of the thymus from immunized IBMI-huNSG mice revealed dynamic adhesive contacts of human-derived thymocytes and DCs accompanied by Rap1 activation. These findings demonstrate that an increase in HLA-DR+ cells by immunization promotes HLA-restricted thymocyte expansion in humanized mice, offering a unique opportunity to generate humanized mice with ease.

## 1. Introduction

Recent progress in the human-to-mouse xenotransplantation of hematopoietic stem cells (HSCs) has made it possible to study human immune development and function in vivo, and provide preclinical models for human infection, autoimmunity, and cancer, as well as the development of vaccines and drugs [1,2,3,4,5]. Highly efficient reconstitution of a human hemato-lymphoid system has been achieved in several mouse strains that are compromised for both innate and adaptive immunity, either spontaneously (NOD SCID) or as a result of genetic engineering (Rag1/2^−/−^, γc^−/−^), resulting in a lack of functional T-, B-, and NK cells; consequently, the mice are tolerant of human grafts. Engraftment of CD34+ HSCs from human cord blood into such immunodeficient mice successfully reconstitutes and maintains the human hemato-lymphoid system over the long term, leading to the establishment of humanized mice; this approach has been used frequently. However, it has become apparent that T-cell functions and humoral immune responses in those mice are severely compromised [6]. These defects have been attributed to the lack of HLA expression in the thymus. Human thymocytes selected by murine MHC and matured in the thymus would be anergic and unable to recognize the antigen presented by HLA+ dendritic cells (DCs) in the periphery. In support of this notion, T- and B-cell function is improved in mice receiving transplants of human fetal thymus and liver together with HSCs from the same donor [7]. The transgenic expression of HLA class I in mice improves HLA-A2–restricted CD8 T-cell responses, although the efficiencies of human T-cell reconstitution and B-cell functions are not affected [8,9,10]. By contrast, transgenic mice expressing HLA class II with or without class I have higher levels of functional human T- and B-cells with immune responses, including antigen-specific antibody production [11,12]. Thus, the interactions of human-derived immature thymocytes with HLA molecules, especially class II expressed in thymic epithelial cells, is crucial for the generation of immuno-competent humanized mice. However, the mechanisms of T-cell development and selection in humanized mice remain obscure. Humanized mice are xenogeneic, and both human and mouse MHC may participate in thymocyte selection. Previous studies suggest that human thymocytes can interact with mouse MHC on thymic stroma, or alternatively with human MHC on developing immature thymocytes or DC-like cells [13,14]. These observations suggest the importance of HLA in the functional thymocyte production of humanized mice. The spatiotemporal expression of human and mouse MHC and their roles in T-cell development in the thymus of humanized mice remain to be determined.

Previously, we reported a novel HTLV-1-infected humanized mouse model generated by the intra-bone marrow injection of human CD133+ stem cells into NOD/Shi-scid/IL-2Rγc null (NOG) mice (IBMI-huNSG mice) [15]. In the infected mice, HTLV-1-specific adaptive immune responses, including neutralizing antibody production, were induced in the absence of a human HLA transgene. These data suggest functional T-cell production in IBMI-huNSG mice in the absence of an HLA transgene. This prompted us to examine the development of T-cells in the thymus of IBMI-huNSG mice. In this study, we found that thymocyte development in IBMI-huNSG mice is controlled by immunization. Immunization promoted single-positive thymocyte production, which was associated with the expansion of the medullary region through the recruitment of human-derived DCs from the periphery. Two-photon imaging of the thymic tissues revealed that human thymocytes interacted dynamically with human DCs, accompanied by integrin-stimulating Rap1 activation at the interface. Because single-positive thymocyte production and peripheral alloreactive T-cell responses are dependent on HLA expression, the thymocyte–HLA interaction is likely critical for functional thymocyte production in IBMI-huNSG mice.

## 2. Results

### 2.1. Development of Lymphocytes in IBMI-huNSG Mice

IBMI-huNSG mice were generated by the intra-bone marrow injection of human CD133^+^ cells into sublethally irradiated NSG mice [15,16]. To monitor the generation of T-cells and B-cells in the IBMI-huNSG mice, we measured human-derived hematopoietic cell subsets in peripheral blood by staining with anti-human CD45, CD3 and CD20 antibodies (Figure 1A,B). Typically, the level of human-derived leukocytes increased, with a peak of 80% chimerity 18 weeks after transplantation, and was maintained over 42 weeks of the monitoring period (Figure 1A). Human B-cells were detected 6 weeks after transplantation, although T-cells were rare at this time point (Figure 1B). Human T-cells were detected later, at 14 weeks, after which their levels increased steadily, as we reported previously [15]. CD4^+^ and CD8^+^ T-cell subsets arose (Figure 1C) that contained naïve (hCD45RA^+^hCD27^+^), memory (hCD45RA^−^CD27^+^), effector memory (hCD45RA^+^hCD27^−^), and effector (hCD45RA^−^hCD27^−^) T-cell subsets, but very few T-cells of the effector phenotype (Figure 1D,E).

We next measured the expression of human CD3, CD4, and CD8 in the thymus of IBMI-huNSG mice (Figure 1F,G). Thymocyte profiles of immature CD4^−^CD8^−^ double-negative (DN), CD4^+^CD8^+^ double-positive (DP), and mature CD4^+^CD8^−^ single-positive (SP), as well as CD4^−^CD8^+^ SP with CD3 expression, were shown. CD3 was upregulated from DN to SP cells, indicative of the normal development of thymocytes in IBMI-huNSG mice.

### 2.2. Immunization Promotes Thymocyte Maturation

Histological examination has revealed that human CD3^hi^ thymocytes were localized in clusters near the capsule of the thymus of IBMI-huNSG (Figure 2A), in contrast to the distribution of mature thymocytes in the inner region (medulla) with immature thymocytes in the outer region (cortex) of the thymus of wild-type mice. The aberrant localization of mature human thymocytes was reported previously in humanized mice [17].

Environmental factors such as inflammation or microflora could the affect establishment of thymic structures [18,19]. Therefore, we hypothesized that immunological stimuli could reorganize thymic structures and promote thymocyte development. To test this idea, IBMI-huNSG mice were immunized with the subcutaneous injection of ovalbumin emulsified with complete Freud’s adjuvant. One week after immunization, IBMI-huNSG mice had enlarged thymus with higher numbers of thymocytes (Figure 2B left). In addition, there was substantial expansion of the medulla-like regions, in which CD3^hi^ CD4^+^ SP and CD3^hi^ CD8^+^ SP cells were localized (Figure 2B right). Immunization also increased the numbers of DP and CD3^+^DP developing thymocytes (Figure 2C,D). The number of CD4 SP thymocytes 2 weeks after immunization was higher than that in unimmunized mice (Figure 2E). The CD69 and CD62L profiles of CD4 SP cells revealed that both semi-mature (CD69^high^CD62L^low^) and mature (CD69^dull^CD62L^high^) subsets of CD4 SP cells [20,21,22] were increased upon immunization (Figure 2F,G). These results indicate that peripheral immunological stimuli promoted the selection and maturation of thymocytes in humanized mice.

### 2.3. Enhanced Human Thymocyte Production Associated with Expansion of Human-Derived Cells Expressing HLA-DR

The selection of thymocytes is elicited by recognition of peptide-MHC on antigen-presenting cells. In humanized mice, it remains unclear whether thymocytes are selected by mouse-derived MHC or human-derived HLA, although several reports have shown that the ectopic expression of HLA-DR promotes functional T- and B-cell responses [11,17,23]. To address this issue, we first examined the expression of mouse and human MHC class II in the thymus by immunohistochemistry. HLA-DR^+^ cells, scattered in the thymus in unimmunized mice, increased dramatically after immunization (Figure 3A). In both unimmunized and immunized IBMI-huNSG mice, human MHC-II^+^ (HLA-DR^+^) cells were segregated from mouse MHC-II^+^ (I-A^g7+^) cells, and colocalized with human CD3^+^ thymocytes (Figure 3A). Flow cytometric analysis showed that HLA-DR^+^ cells in the thymus from immunized IBMI-huNSG mice were present in the hCD45^+^ hematopoietic population, with a few HLA-DR^+^ cells in the hCD45^−^mCD45^−^ population, whereas I-A^g7^ was expressed in non-hematopoietic cells (hCD45^−^mCD45^−^) (Figure 3B), indicating that human-derived hematopoietic cells contributed predominantly to the HLA-DR^+^ population (Figure 3B). The number of HLA-DR^+^ cells increased considerably after immunization (Figure 3C,D). Furthermore, HLA-DR^+^ cells morphologically resembled DCs with long dendrites, and also expressed the DC marker CD205 in the medulla-like regions (Figure 3E). Together, these results indicate that HLA-DR^+^ hematopoietic cells including DCs are increased in enlarged medulla-like regions after immunization, suggesting an important role for human-derived cells expressing HLA-DR in promoting thymocyte selection and maturation in immunized IBMI-huNSG mice.

### 2.4. Human Thymocyte Migration and Interactions with DC

Thymocyte development and maturation occur in distinct regions and are closely associated with cellular motility [24]. In both mice and humans, immature thymocytes acquire high trafficking capabilities as they mature and move from the cortex to the medulla [14,25,26,27]. To visualize human thymocyte motilities in the thymus of IBMI-huNSG, we lentivirally introduced the fluorescent Venus protein into CD133^+^ cord- blood HSCs before transferring them into NSG mice. Explanted thymic lobes were subjected to two-photon imaging. The cortex, medulla and cortico-medullary junction of the thymus were identified with collagen signals visualized with second harmonic generation and visual inspection. Distinct thymocyte motilities were clearly observed in the thymus of IBMI-huNSG mice (Figure 4A, Appendix A). In the cortex-like region, thymocytes adopted a round shape and crawled randomly (Figure 4B left, Appendix A). By contrast, thymocytes were elongated and migrated more rapidly in the medulla-like region, which was demarcated from the cortex-like region by a collagen-rich junctional zone shown in Figure 4B right and Appendix A. Thymocytes frequently traveled through the narrow gaps in the junctional zone (Figure 4B middle, Appendix A). The two-dimensional plot of velocity and meandering index calculated from trajectories of thymocytes in each region reveals that thymocytes moved faster and more directionally in the medulla-like regions (Figure 4B,C). Consistent with this, cortex thymocytes, which adopted a circular shape, became elongated in the junctional and medulla-like regions (Figure 4D).

The small GTPase Rap1 is a strong integrin activator that induces adhesion triggered by chemokines and antigens [28,29]. Activated Rap1-GTP accumulates in the immunological synapses (IS) of T-cell–APC interactions [30], which is also observed with mature thymocytes [31]. To investigate whether human-derived thymocytes and DCs interacted functionally, we lentivirally introduced a Rap affinity probe (GFP-RalGDS-RBD) into CD133^+^ cord blood HSCs to detect Rap1 activation, and examined thymocyte migration and interactions in the thymus of immunized IBMI-huNSG mice. Although both thymocytes and DCs are visualized by this method, DCs can be easily morphologically distinguished from thymocytes (Figure 4E, Appendix A), as reported previously [13]. When migrating thymocytes stopped and contacted DCs, the Rap1 affinity probe accumulated in contact areas, indicative of Rap1 activation upon T-cell–DC interactions (Figure 4E). Rap1 activation was sustained during T-cell–DC interactions, as reported in mouse thymocyte IS [31]. The average contact duration between thymocytes and DCs was approximately 2 min (Figure 4F), classified as transient contact, as frequently observed in positively selected thymic environments [32,33]. Together, these observations suggest that the brief contacts in human thymocyte–DC interactions provide HLA-triggering TCR signaling that promotes the positive selection of thymocytes in the thymus of IBMI-huNSG mice.

### 2.5. HLA-DR-Dependent Thymocyte Maturation and Allogeneic Response of T-Cells

To determine whether the recognition of HLA-DR by TCR facilitates human thymocyte maturation, we isolated thymic lobes from immunized or unimmunized IBMI-huNSG mice for organ culture and incubated them for 4 days in an IL-2-containing medium with high oxygen (Figure 5A). The generation of human CD4 SP cells was substantially higher in thymic lobes derived from immunized mice than in those derived from unimmunized mice (Figure 5B), indicating that immunization facilitates the production of mature CD4^+^ thymocytes. To determine the impact of HLA-DR blockade on the thymocyte maturation, we cultured thymic lobes from immunized mice in the presence or absence of anti-HLA-DR antibodies. Although the majority of thymocytes became CD4 or CD8 SP cells on day 4 of the culture, the addition of anti-HLA-DR antibodies decreased the generation of CD4 SP cells without affecting CD8 SP or DP cells (Figure 5C,D). These observations indicate that CD4 SP thymocytes from immunized IBMI-huNSG mice were selected and matured by HLA-DR.

If thymocytes are appropriately selected by HLA, peripheral mature T-cells should be activated in an HLA-dependent manner. To investigate this issue, we measured the alloreactive T-cell response against human-derived cells (or mouse-derived cells, as a control). T-cells from IBMI-huNSG were co-cultured with human PBMCs or splenocytes from C57BL/6 mice (H-2^d^) (Figure 5E). Human T-cells from immunized IBMI-huNSG did not respond to mouse splenocytes and syngeneic huNSG bone marrow cells, but did proliferate in response to human PBMCs, supporting the notion that T-cells from IBMI-huNSG mice were selected by HLA. Collectively, these data suggest that immunization promotes the functional positive selection of thymocytes to generate HLA-restricted mature T-cells.

## 3. Discussion

The direct injection of CD133^+^ cord-blood HSCs into the bone marrow of NSG mice reconstitutes human T-cell and B-cells with high chimerity [15]. In addition, myeloid-lineage cells including macrophage and monocytes are also developed efficiently in these mice [34,35]. In such humanized mice, thymocytes develop normally, but the thymus exhibits aberrant tissue organization and thymocyte distribution [36]. We showed that the immunization of IBMI-huNSG mice increased the numbers of developing DP, and subsequently both semi-mature and mature SP thymocyte subsets in the thymus. The expansion of thymocytes was associated with the drastic expansion of the medulla-like area enriched in HLA-DR+ APCs, such as CD205+ DC cells. In addition, thymic organ culture experiments revealed that increased CD4 SP thymocyte production was HLA class II-dependent. Consistent with this, peripheral human T-cells from immunized IBMI-huNSG mice exhibited HLA-restricted allogeneic responses, indicating that immunization promoted the positive selection and maturation of thymocytes. Thus, the increase in human APC in the thymus is essential for the immunization-induced enhancement of mature T-cell production. Our findings agree with previous studies showing the importance of transgenic expressions of HLA class I and class II for functional T-cell responses in humanized mice [8,9,11], and further elucidate the cellular mechanism of HLA-driven functional T-cell generation in the absence of HLA transgenes. Collectively, our results support the conclusion that thymocyte expansion and differentiation occurred in humanized mice after immunization. Of note, we could not exclude another possibility that recirculating T-cells are recruited to the thymus and form ectopic secondary lymphoid structures, which are not mutually exclusive.

The imaging of thymocytes within thymic tissues revealed that the migration patterns of human thymocytes in IBMI-huNSG mice are similar to those of human and murine thymocytes [14,25,26,27]. In the normal thymic environment, immature DP cells in the cortex exhibit random and slow crawling; by contrast, in the transition to SP cells facilitated by positive selection, thymocytes greatly enhance their trafficking capability, exhibiting a rapid and directional migration in the medulla. We found that in the thymus from IBMI-huNSG mice, thymocyte motilities were upregulated from the cortex to medulla. In the HLA^+^ DC-rich medulla-like regions, human thymocytes transiently interacted with human DCs. A previous study reported thymocyte–DC contacts or homotypic thymocyte contacts within the thymus of humanized mice [13]; however, it remains unclear whether these interactions are functional events. The introduction of the Rap1 affinity probe visualized Rap1 activation and accumulation upon contact of human thymocytes and DCs, indicating TCR-triggering and integrin-dependent contacts between thymocytes and DCs.

Because HLA was almost exclusively expressed in enlarged medulla-like areas of the thymus in IBMI-huNSG mice, both the positive and negative selection of human thymocytes by HLA should occur in medulla-like areas. In mouse models, peripheral DCs migrate to the thymus and participate in the induction of central tolerance against peripheral antigens [37,38,39]. Thus, inflammatory signals from immune responses reorganize the thymic environment by recruiting human-derived DCs and other APCs from the periphery. Alternatively, the inflammatory signals induce the proliferation of APCs within the thymus. The expansion of HLA-DR+ APCs promotes the selection of human thymocytes to supply functional T-cells. Two-photon imaging of the thymus from immunized humanized mice revealed that most thymocyte–DC contacts were transient, with an average contact duration of 2 min, which is much shorter than the contacts in a negatively selecting environment [25,33]. Thus, the transient thymocyte–DC interactions observed under our conditions likely generate TCR signaling that favors positive selection but is not sufficient for negative selection.

In summary, immunization signals expand human HLA-expressing cells in the thymus of IBMI-huNSG mice, and promote HLA-dependent thymocyte expansion. Further studies are required to elucidate the mechanisms of HLA-DR+ APC expansion and thymocyte selection linked with the immune response in IBMI-huNSG mice. Our findings reveal a strategy for generating humanized mice to evaluate efficient human immune responses without requiring HLA matching, as in HLA-transgenic humanized mouse systems.

## 4. Materials and Methods

### 4.1. Animals and Cells

NSG mice (NOD. Cg-*Prkdc^scid^ Il2g^tm1wjl^*/SzJ) were purchased from Charles River Japan and maintained in the specific pathogen free condition in the animal facility of Kansai Medical University. All animal experiments were approved by the Animal Care Committees of Kansai Medical University. Human cord blood was obtained from the Japan Red Cross cord blood bank and Riken Bioresource Research Center (Kyoto, Japan) with documentation of informed consent.

### 4.2. Antibodies

FITC-, PE-, PE-Cy7-, PE-Cy5-, APC-, APC-Cy7-, or eFluor 450-labeled monoclonal antibodies (Ab) specific for human (h) CD3 (PE, clone: UCHT1), hCD4 (APC, clone: RPA-T4), hCD8 (eFluor 450, clone: RPA-T8), hCD20 (PE-Cy7, clone: 2H7), hCD27 (PE-Cy5, clone: 0323), hCD45 (APC-Cy7, clone: HI30), hCD69 (PE-Cy7, clone: FN50), and HLA-DR (APC, clone: L243) were obtained from Biolegend (San Diego, CA, USA). Monoclonal Ab for hCD62L (FITC, clone: DREG-56), hCD45RA (FITC, clone: HI100), mouse (m) CD45.1 (PE-Cy7, clone: A20), and I-A^k^ (FITC, clone: 10-3.6), which cross-reacts with I-A^g7^ of NOD mice, was purchased from eBioscience (San Diego, CA, USA).

### 4.3. Generation of IBMI-huNSG Mice

The generation of IBMI-huNSG mice has been described previously [15,16]. In brief, CD133^+^ cells were isolated from cord blood using an AutoMACS Separator (Miltenyi Biotec, Bergisch Gladbach, Germany). In total, 5 × 10^4^ CD133^+^ cells were intrafemorally injected into each mouse under anesthesia after 2.5 Gy sublethal γ-ray irradiation.

### 4.4. Flow Cytometric Analysis

Thymocytes and peripheral leukocytes in IBMI-huNSG mice were isolated and stained with fluorescence-labeled antibodies specific for T-cells, B-cells, or myeloid cells, as indicated. Stained cells were measured with FACS Canto II (BD Bioscience, San Jose, CA, USA). The data were analyzed using the FlowJo V10 software (FlowJo, Ashland, OR, USA).

### 4.5. Immunization

Mice were subcutaneously immunized with trinitrophenyl (TNP)-conjugated ovalbumin (OVA) emulsified with complete Freud’s adjuvant. After 1–2 weeks of immunization, immunized and unimmunized paired mice were sacrificed and subjected to analysis.

### 4.6. Lentiviral Vector Transfection

To visualize thymocytes and Rap1 activation, a fluorescent Venus protein or Rap affinity probe (GFP-RalGDS-RBD) was cloned into the lentiviral vector CSII-UBC-MCS as described [31]. Packaged lentiviruses were transduced into human CD133^+^ cells by exposure to the packaged viruses at a concentration corresponding to a multiplicity of infection (MOI) of 50–100. The transduced cells were cultured for 24 h in IMDM medium containing 4% FCS, 100 ng/mL stem cell factor, 20 ng/mL thrombopoietin, and 100 ng/mL Flt3 ligand, and used for transplantation.

### 4.7. Immunohistochemistry

Frozen sections fixed with acetone/methanol (1:1) were stained with 5–10 mg/mL antibody against hCD3, hCD4, hCD8, hCD205, or HLA-DR, or mouse MHC class II I-A^k^ (10-3.6) Ab. DAPI (1 mg/mL) was used for counterstaining, and imaging was performed on a confocal microscope (LSM 510 META, LSM 700, Zeiss, Clackamas, OR, USA). Images were analyzed using the ZEN 2 software (Zeiss).

### 4.8. Imaging of Thymic Tissues Using Two-Photon Microscopy

The live imaging of thymic tissues was performed as described previously with some modification [40]. In short, the isolated thymus was cut in half on a Vibratome (Dosaka EM, Kyoto, Japan) and perfused with warm IMDM medium containing 4% FCS oxygenized with 95% oxygen/5% carbon dioxide. Images were acquired on a Fluoview FV1000MPE system (Olympus, Tokyo, Japan), with a Ti:sapphire laser (MaiTai HP DeepSee-OL, Spectra-Physics, Milpitas, CA, USA) tuned to 890 nm. The acquired images were analyzed using the Volocity 6.3 3D software (PerkinElmer, Waltham, MA, USA).

### 4.9. Thymic Organ Culture

Isolated thymic lobes from IBMI-huNSG mice were introduced into 24-well plates containing IMDM medium with 4% FCS and 10 ng/mL hIL-2 in the presence of anti–human HLA-DR neutralizing antibodies or isotype control antibodies. Thymic lobes were cultured at 37 °C in a sealed bag packed with gas consisting of 95% O_2_ and 5% CO_2_. Cultured thymic lobes were harvested after 4 days and subjected to flowcytometric analysis.

### 4.10. Mixed Lymphocyte Reaction

T-cells were isolated from the spleen of humanized mice using a negative selection kit (RosetteSep; STEMCELL Technologies, Vancouver, BC, Canada). Syngeneic stimulator cells were obtained from bone marrow cells derived from the same humanized mice. Allogeneic stimulator cells from human PBMCs were obtained from blood of a healthy control. Splenocytes from C57BL/6 mice were used as xenogeneic antigen-presenting cells. Stimulator cells were exposed to 30 Gy γ-irradiation. After 72 h, a mixture of each type of stimulator cell and 1 × 10^5^ T-cells from humanized mice was incubated in IMDM medium containing 4% FCS. [^3^H]-thymidine was added to each well, and the sample was incubated for an additional 8 h before harvest and measurement using a scintillation counter.

### 4.11. Statistical Analysis

Statistical analysis was performed using GraphPad Prism 8 (GraphPad, Boston, MA, USA) and the Excel software 2019 (Microsoft, Redmond, WA, USA).

## Figures and Tables

**Figure 1 ijms-24-11705-f001:**
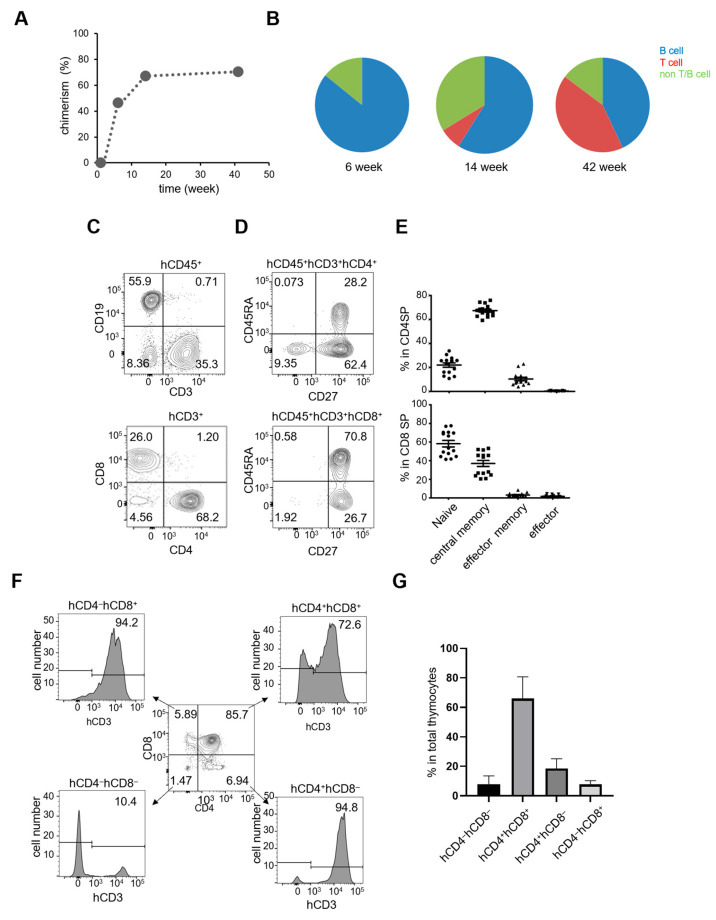
Generation of lymphocytes in IBMI-huNSG mice. (**A**) Typical time course of chimerism of human hematopoietic cells (human (h) CD45^+^ and mouse (m) CD45^−^ cells) in the peripheral blood of IBMI-huNSG mice. (**B**) The proportion of B-cells (blue), T-cells (red), and non-T/B-cells (green) in hCD45^+^ cells in IBMI-huNSG mice. Weeks post-transplantation are indicated. (**C**) Representative flow cytometric profiles of human-derived peripheral T-cells (hCD3^+^) and B-cells (hCD19^+^) in IBMI-huNSG mice (14 weeks post-transplantation) (upper). Human CD4^+^ (hCD4^+^) and CD8^+^ T-cells (hCD8^+^) are also shown (below). (**D**) Flow cytometric profiles of hCD27 and hCD45 in CD4^+^ and CD8^+^ T-cells. (**E**) Fractions of naïve (hCD45RA^+^hCD27^+^, closed circle), memory (hCD45RA^−^hCD27^+^, closed square), effector memory (hCD45RA^−^hCD27^−^, closed triangle), and terminal differentiated effector (hCD45RA^+^hCD27^−^, closed inverted triangle) cells in CD4^+^ (upper) and CD8^+^ T-cells (lower). The bars represent average percentages (*n* = 14). (**F**) Thymocyte development in IBMI-huNSG mice. Expression of CD3 in DN (hCD4^−^hCD8^−^), DP (hCD4^+^hCD8^+^), CD4 SP (hCD4^+^hCD8^−^), and CD8 SP (hCD4^−^hCD8^+^) thymocytes of IBMI-huNSG mice (14 weeks). (**G**) The frequency of thymocyte subsets of IBMI-huNSG mice (*n* = 10).

**Figure 2 ijms-24-11705-f002:**
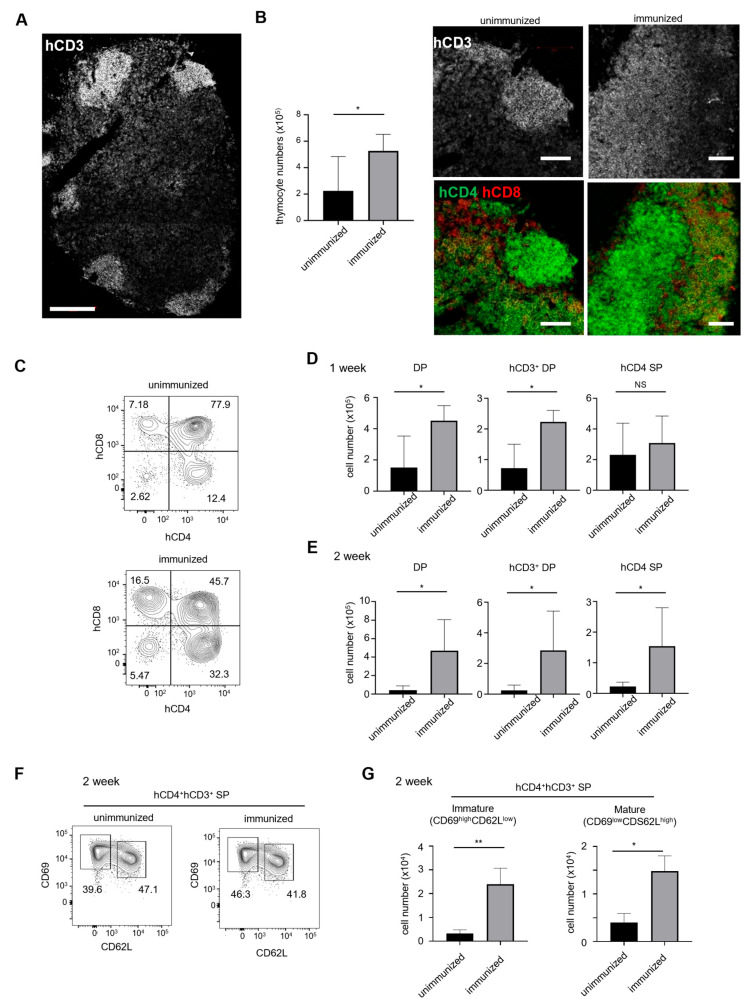
Enhanced thymocyte development in enlarged medulla-like regions of the thymus from humanized mice after immunization. (**A**) Representative images of the thymus stained with anti-hCD3 antibody show focal aggregation of developed thymocyte near the capsule in IBMI-huNSG mice (28 weeks). Bar, 200 μm. (**B**) Total thymocyte numbers before and 1 week after immunization (left). Averages with SEM are shown (unimmunized, *n* = 5; immunized, *n* = 5). Representative images of the thymus from unimmunized and immunized IBMI-huNSG mice stained with antibodies against hCD3, hCD4 (green), and hCD8 (red). Bars represent 100 μm. (**C**) Representative flowcytometric profiles of hCD4 and hCD8 in the thymus from unimmunized and immunized IBMI-huNSG mice (day 7). (**D**,**E**) Changes in cell numbers of DP, CD3^+^ DP, and CD4 SP thymocytes with or without immunization. Cell numbers 1 week (unimmunized, *n* = 5; immunized, *n* = 5) (**D**) and 2 weeks after immunization (unimmunized, *n* = 5; immunized, *n* = 6) (**E**) are shown. Averages with SEM are shown. (**F**) hCD69 and hCD62L expression profiles of hCD4 SP cells measured by flow cytometry. The gate strategy and percentage of CD69^high^CD62L^low^ and CD69^low^CD62L^high^ are shown and represent immature and mature subsets, respectively. (**G**) Changes in cell numbers of CD69^high^CD62L^low^ and CD69^low^CD62L^high^ CD4 SP thymocytes with or without immunization. Cell numbers at 2 weeks (unimmunized, *n* = 5; immunized, *n* = 6). The statistical significance of the above data (**D**,**E**,**G**) was calculated by Student’s *t*-test. * *p* < 0.05, ** *p* < 0.01, NS: not significant.

**Figure 3 ijms-24-11705-f003:**
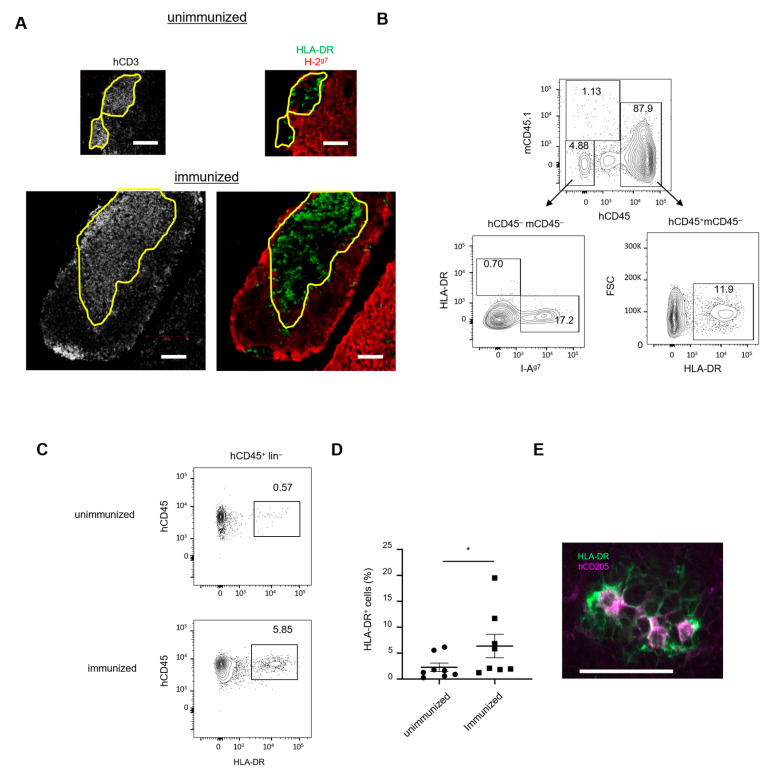
Recruitment of HLA-DR^+^ DCs in the thymus after immunization. (**A**) Distribution of human HLA-DR and mouse MHC class II in the thymus. Frozen sectioned thymic tissues from immunized IBMI-huNSG mice were stained with antibodies against hCD3 (left images) or HLA-DR (green) and I-A^g7^ (red) (right images). Bars represent 100 μm. (**B**) Flow cytometric profiles of hematopoietic or non-hematopoietic cells in the thymus from IBMI-huNSG mice (upper). Non-hematopoietic cells gated for hCD45^−^ mCD45.1^−^ contained the I-A^g7+^ population (17.2%) but few HLA-DR^+^ cells (0.7%) (lower left), whereas human-derived hematopoietic cells (hCD45^+^ mCD45^−^) contained the HLA-DR^+^ population (11.9%) (lower right). (**C**) HLA-DR expression in the human hematopoietic cells negative for the lineage markers hCD3 (T-cells), hCD20 (B-cells), hCD14 (monocytes/macrophages), and hCD56 (NK cells), with/without immunization. (**D**) Percentages of the HLA-DR^+^ in hCD45^+^lin^−^ cells with/without immunization (unimmunized, *n* = 8; immunized, *n* = 8). Bars represent averages with SEM of the populations. Statistical significance was determined by Mann–Whitney U test (* *p* < 0.05). (**E**) Immunostaining for HLA-DR (green) and CD205 (red) in thymus from immunized IBMI-huNSG mice. Bar represents 50 μm.

**Figure 4 ijms-24-11705-f004:**
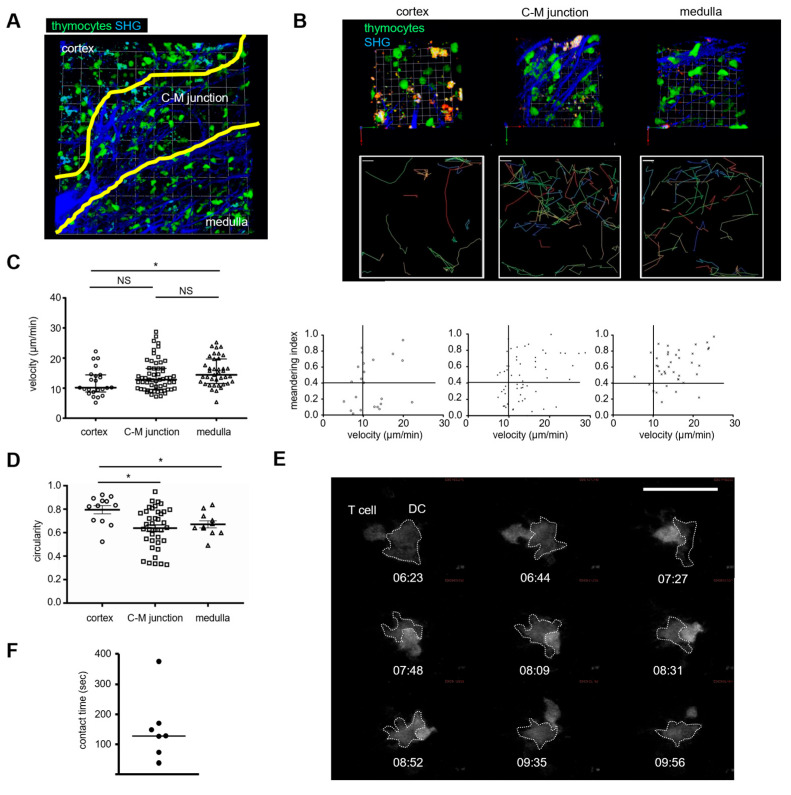
Two-photon imaging of human thymocyte–DC interactions in the thymus. (**A**) Two-photon live imaging of the thymus from IBMI-huNSG mice transplanted with CD133^+^ HSCs expressing Venus. Most Venus-expressing cells are morphologically identified as thymocytes and DCs. Cortex, medulla, and cortico-medullary junction rich in fibrous collagen (blue) are indicated. One square unit represents 25 μm × 25 μm. The data are representative of four experiments. (**B**) Comparison of cell motilities of thymocytes in the cortex, cortico-medullary junction, and medulla. Representative 3D reconstituted images of each region of the thymus (upper, 1 × 1 unit = 7.5 μm × 7.5 μm) and trajectories of thymocytes (middle, a scale bar, 5.1 μm) are shown. The plot of thymocyte velocities (μm/min) and meandering index of trajectories (straightness) in each region is also shown (bottom). (**C**) Thymocyte velocities (μm/min) in the cortex, cortico-medullary junction, and medulla. (**D**) Comparison of thymocyte circularity in the cortex, cortico-medullary junction, and medulla. Bars represent averages and SEM. (**E**) A representative image sequence of thymocyte–DC interactions is shown with time (min: s). Thymocyte and DCs (marked with dashed lines) express a Rap affinity probe. Bar represents 30 μm. The data are representative of two experiments. (**F**) Contact duration of thymocyte–DC interactions (*n* = 7). Statistical significance above data (**C**,**D**) by unpaired Student *t*-test are indicated: * *p* < 0.05, NS: not significant).

**Figure 5 ijms-24-11705-f005:**
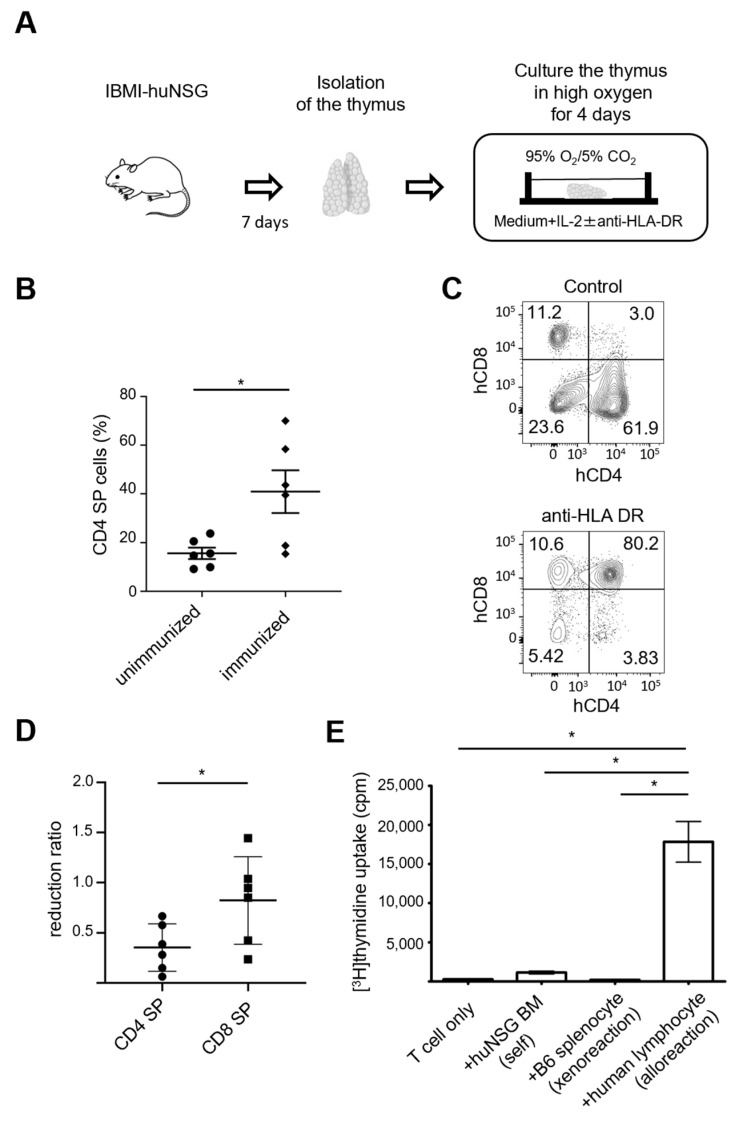
HLA-DR-dependent thymocyte maturation and allogeneic response of T-cells. (**A**) Schema of organ culture of the thymus. Thymic lobes were isolated 7 days after immunization of IBMI-huNSG mice and cultured in the absence or presence of anti-HLA-DR antibodies. Organ culture was supplemented with IL-2 and high oxygen. (**B**) Percentages of CD4 SP thymocytes in cultured thymus from humanized mice with/without immunization (unimmunized, *n* = 6; immunized, *n* = 6). Bars are averages with SEM. (**C**,**D**) Anti-HLA-DR antibody inhibited the generation of CD4 SP thymocytes. Representative flow cytometric profile (**C**) and reduction ratios of CD4 and CD8 SP thymocytes normalized against control antibodies (unimmunized, *n* = 6; immunized, *n* = 6). Bars represent averages with SEM of the populations. (**E**) Mixed lymphocyte reaction of peripheral T-cells from immunized IBMI-huNSG mice using splenic T-cells from syngeneic bone marrow cells from huNSG mice (huNSG BM), xenogeneic splenocytes from C57BL/6 mice, and allogeneic human-derived peripheral lymphocytes. Proliferation, as reflected by the uptake of [^3^H]-thymidine measured in triplicate (day 3), is shown. Averages with SEM are shown, with statistical significance (unpaired Student *t*-test; * *p* < 0.05). The data are representative of two experiments. Statistical significance above data (**B**,**D**,**E**) was determined by paired *t*-test: * *p* < 0.05.

## Data Availability

All data generated or analyzed included in the current manuscript are available from the corresponding author upon request.

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
