# Peer review of "Thymocyte Development of Humanized Mice Is Promoted by Interactions with Human-Derived Antigen Presenting Cells upon Immunization"

_ijms, 2023, doi:10.3390/ijms241411705_

Round 1

Reviewer 1 Report

Fukuhara et al claim that thymopoiesis in humanized mice is promoted by interaction with DCs, following immunization. The authors humanize NSG mice by intra-femur injection of CD133-purified human cord blood cells, following sub-lethal irradiation. The authors state that the CD4/CD8 profile of humanized mice is normal despite an abnormal thymus structure and proceed with characterization of the humanized thymi using FACS and microscopy of thymus slices to image live thymocytes in humanized thymic explants. The data is of interest but does not support the main conclusion stated in the title. I suggest a major reformulation of the text, including a tone down of the title or adding the experiments missing that could come to support the authors’ claims.

General points:

One major aspect that is unclear is whether the human CD3-high cells in the thymus, specifically in the aberrant thymic regions under the capsule, are indeed CD4 and CD8 single positive thymocytes or whether they could be circulating T cells, homing to an ectopic, secondary lymphoid structure-like that rather than a thymic medulla. To this aspect, it would be important to stain the epithelium and determine whether there are medullary thymic epithelial cells in these regions and cortical thymic epithelial cells in the cortex.

Still in this context, the authors should clarify whether upon immunization thymopoiesis is indeed affected or whether circulating T cells are simply proliferating. The timing seems more consistent with T cell expansion than an effect on thymopoiesis. The authors should measure proliferation in the cells and add markers that permit distinguishing thymocytes from circulating T cells.

Also relevant is the quantification and characterization of the human dendritic cells that the authors consider so important. Are these cells increased in absolute numbers? Assuming so, does that result from recruitment or rather proliferation of existing cells of the thymus?

Specific points:

To Figure 1:

i) The authors show that the CD4/CD8 profile of humanized mice is normal but give one single example. The authors should quantify percentages and numbers in several mice.

ii) Please indicate which organ is shown in C-E in the legend

To Figure 2:

i) representative sections of the whole thymus should be shown for CD4/CD8 as shown in A) to permit assessing thymus structure. The image should be larger and have better resolution. Double positive and single positive thymocytes should be clearly visible. The authors should also stain for DC and TEC markers in adjacent thymus sections and/or by combining thymocyte and stromal markers. How many thymi did the author analyze histologically? Add statistics.

ii) The conclusion “These results indicate that peripheral immunological stimuli promoted selection and mat-204uration of thymocytes in humanized mice.” Is not supported by the data shown.

To Figure 3:

i) The Increase in T cells following immunization could be explained by T cell expansion induced by DC stimulation (also taking into account that it occurs within a 1wk time frame) rather than an impact on thymopoiesis.

ii) Please characterize these regions for markers that show unequivocally that they are medulla like, or whether they instead resemble more a secondary lymphoid organ.

iii) The increase in the % of HLA-DR-positive cells (presumably DCs) could be both explained by recruitment, like the authors state in the figure title, or proliferation in the thymus. Either way, quantification would require not only determining % but also absolute numbers in the organ. In D) % refers to what (ie percentage of cells within what population)? It is unlikely that DCs reach the indicated % in total cells, but the authors need to specify.

iv) In E) HLA-DR and CD205 co-expression should be quantified. In addition, the authors should show and/or discuss why do they increase in immunized mice. The authors state in the Figure title that DCs are recruited but do not show that. There should be a quantification of proliferation in DCs in the humanized thymi and an attempt to address whether the celld are indeed recruited to the thymus or whether thymic DCs proliferate in response to immunization.

v) In the text (lines 270-271) the authors should correct the text. As is, it seems like they consider possible that thymic epithelial cells could be generated from injected human hematopoietic progenitors.

To Figure 4:

i) Figure 4 is cut to the right. I don’t understand how do the authors define C-M junction: it should be a line and rather than a region...

ii) Authors should indicate how many thymi were analyzed and add statistics

iii) In E) the authors show cell interactions, presumably between DCs and thymocytes. What stage is imaged? DP or SP? If SP, how can the authors distinguish between SP thymocytes and circulating T cells? Markers should be added to the staining s to support (or not) the author’s claim. Neither the figure nor the video is conclusive in this regard

v) The conclusion “Together, these observations suggest that the brief contacts in human thymocyte-DC interactions provide HLA-triggering TCR signaling that promotes positive selection of thymocytes in the thymus of IBMI-huNSG mice.” Is not supported by the data shown.

To Figure 5:

i) Authors should indicate how many thymi were analyzed and statistics

ii) I repeat that it is unclear that thymopoiesis is indeed affected, and that these regions are indeed medulla-like rather than ectopic secondary lymphoid tissues promoting cell stimulation and expansion rather than thymopoiesis.

iii) The statement “Collectively, these data suggest that immunization promotes functional positive selection of thymocytes to generate HLA-restricted mature T cells.” Is not supported by the data.

No further comments

Reviewer 2 Report

Title: Thymocyte development of humanized mice is promoted by interactions with human-derived dendritic cells upon immunization.

On this manuscript the authors investigated thymocytes development in IBMI-huNSG mice with special interest in the role of human and mouse MHC. They also analyze thymic structure and the interaction of human-derived thymocytes and DC demonstrating that recruitment of human DC by immunization promotes HLA-restricted thymocytes development in humanized mice.

This reviewer suggest the following minor changes:

1. Line 46. Add the word "functional" after "in lack". There is not total absence of these cell populations. Although, some of these cells are present, their function is affected.

2. Line 90. Should have a sentence a log the lines such as: "All animal experiments were approved by the Animal Care Committees of Kansai Medical University. All mice were handled in sterile conditions and maintained in germ-free isolators located in……"

3. Lines 92-102. Each antibody should match its own fluorochrome. e.g. hCD3 FITC. In addition, this section should go into "flow cytometry analysis" section and "flow cytometry analysis should go after "generation of IBMI-huNSG mice".

4. Line 97. Please use hCD3 instead of CD3.

5. Line 106. Reference (15) do not describe the intra-bone marrow injection method. I suggest to use instead the following reference: J Wang, T Kimura, R Asada, et al. SCID-repopulating cell activity of human cord blood-derived CD34- cells assured by intra-bone marrow injection. Blood, 101 (8) (2003), pp. 2924-2931.

6. Line 107. Please delete "More than".

7. Line 171. These two papers do not describe the methods. Please use J Wang et al.

8. Line 181. Please describe the effector phenotype (hCD45RA- hCD27-).

9. Please add hCD3 (yellow) as in figure 3.

10. Figures 4B and 4E are incomplete shown in the manuscript.

11. Lines 302 to 311 have different font sizes.

12. Line 349. Please delete space to join with line 350.

13. Lines 353 to 359 have different font sizes.

14. Line 363. Please add a space between CD4 and SP.

15. Line 391. Please add a space between CD4 and SP

Author Response

We appreciate the reviewer’s comment to improve our manuscript.

Point by point responses.

  1. Line 46. Add the word "functional" after "in lack". There is not total absence of these cell populations. Although, some of these cells are present, their function is affected.

We add “functional after “in lack of”.

  1. Line 90. Should have a sentence a log the lines such as: "All animal experiments were approved by the Animal Care Committees of Kansai Medical University. All mice were handled in sterile conditions and maintained in germ-free isolators located in……"

We added the statement of Animal Care and condition in Animal section.

  1. Lines 92-102. Each antibody should match its own fluorochrome. e.g. hCD3 FITC. In addition, this section should go into "flow cytometry analysis" section and "flow cytometry analysis should go after "generation of IBMI-huNSG mice".

Antibodies section contained the description of not only antibodies for flow cytometry but also antibodies for immunostaining. Thus, we would like to leave ”Antibody section” independently. We moved “Flow cytometric analysis” section after “Generation of IBMI-huNSG mice” section.

  1. Line 97. Please use hCD3 instead of CD3.

 We have described as “human (h) CD3” because of the first abbreviation.

  1. Line 106. Reference (15) do not describe the intra-bone marrow injection method. I suggest to use instead the following reference: J Wang, T Kimura, R Asada, et al. SCID-repopulating cell activity of human cord blood-derived CD34- cells assured by intra-bone marrow injection. Blood, 101 (8) (2003), pp. 2924-293

As the reviewer suggested, we added “J Wang et al 2003”. We leaved Tezuka et al. 2014 for the reference of transplantation using CD133+ HSC.

  1. Line 107. Please delete "More than".

As the reviewer suggested, we deleted “More than”.

  1. Line 171. These two papers do not describe the methods. Please use J Wang et al.

As the reviewer suggested, we added “J Wang et al 2003”. We leaved Tezuka et al. 2014 for thereference of transplantation using CD133+ HSC.

  1. Line 181. Please describe the effector phenotype (hCD45RA- hCD27-).

As reviewer suggested, we described the effector phenotype (hCD45RA- hCD27-).

  1. Please add hCD3 (yellow) as in figure 3.

We added hCD3 in Figure 3.

  1. Figures 4B and 4E are incomplete shown in the manuscript.

We corrected the size of the figure 4 to show Figures 4B and 4E

  1. Lines 302 to 311 have different font sizes.

We corrected font sizes.

  1. Line 349. Please delete space to join with line 350.

We deleted a space.

  1. Lines 353 to 359 have different font sizes.

We corrected font sizes.

  1. Line 363. Please add a space between CD4 and SP.

We added a space.

  1. Line 391. Please add a space between CD4 and SP

We added a space. 

Round 2

Reviewer 1 Report

The authors addressed properly all issues raised and I agree with the acceptance of the paper in the current form.